# Mechanisms of DNA Damage Tolerance: Post-Translational Regulation of PCNA

**DOI:** 10.3390/genes10010010

**Published:** 2018-12-24

**Authors:** Wendy Leung, Ryan M. Baxley, George-Lucian Moldovan, Anja-Katrin Bielinsky

**Affiliations:** 1Department of Biochemistry, Molecular Biology and Biophysics, University of Minnesota, Minneapolis, MN 55455, USA; leung086@umn.edu (W.L.); baxle002@umn.edu (R.M.B.); 2Department of Biochemistry and Molecular Biology, The Pennsylvania State University College of Medicine, Hershey, PA 17033, USA; gmoldovan@pennstatehealth.psu.edu

**Keywords:** PCNA, DNA damage tolerance, translesion synthesis, template switching, ubiquitination, replication stress

## Abstract

DNA damage is a constant source of stress challenging genomic integrity. To ensure faithful duplication of our genomes, mechanisms have evolved to deal with damage encountered during replication. One such mechanism is referred to as DNA damage tolerance (DDT). DDT allows for replication to continue in the presence of a DNA lesion by promoting damage bypass. Two major DDT pathways exist: error-prone translesion synthesis (TLS) and error-free template switching (TS). TLS recruits low-fidelity DNA polymerases to directly replicate across the damaged template, whereas TS uses the nascent sister chromatid as a template for bypass. Both pathways must be tightly controlled to prevent the accumulation of mutations that can occur from the dysregulation of DDT proteins. A key regulator of error-prone versus error-free DDT is the replication clamp, proliferating cell nuclear antigen (PCNA). Post-translational modifications (PTMs) of PCNA, mainly by ubiquitin and SUMO (small ubiquitin-like modifier), play a critical role in DDT. In this review, we will discuss the different types of PTMs of PCNA and how they regulate DDT in response to replication stress. We will also cover the roles of PCNA PTMs in lagging strand synthesis, meiotic recombination, as well as somatic hypermutation and class switch recombination.

## 1. Introduction

Accurate DNA replication is essential for genome stability and cell homeostasis [1,2]. DNA damage encountered during replication poses a continuous threat to genomic integrity. Damage can arise from products of cellular metabolism, such as reactive oxygen species as well as from exogenous sources, such as ultraviolet (UV) and ionizing radiation (IR) [3]. If DNA is left unrepaired, the generation of chromosomal deletions, translocations or complete loss of chromosomes can occur. These alterations of the genome are the major driving force for cancer development [4,5,6,7]. To preserve genome integrity, cells rely on a global DNA damage response (DDR) network to sense and repair DNA damage [8]. Although DDR is highly efficient, some DNA lesions may escape repair and interfere with the progression of replication forks. In this scenario, cells utilize DNA damage tolerance (DDT) pathways to bypass lesions encountered during replication. Lesions will be repaired at a later timepoint, reducing the frequency of stalled replication forks and the risk of fork collapse [9,10,11].

DDT pathways can be subdivided into two branches: translesion synthesis (TLS) and template switching (TS). TLS is catalyzed by specialized low-fidelity DNA polymerases (Pol η, Pol ι, Pol κ, and Rev1) to bypass DNA lesions [12,13,14]. Due to a large active site and the lack of proofreading activity, these polymerases allow for the incorporation of a nucleotide opposite to a damaged DNA template [15]. On the contrary, TS is proposed to use a recombination-like mechanism by which the nascent DNA of the sister chromatid is utilized as a temporary template for replication [16]. The choice between DDT pathways is important because it can determine an error-prone or error-free outcome. A key player driving the choice between TLS and TS is proliferating cell nuclear antigen (PCNA) [17,18].

PCNA is an evolutionarily well-conserved homotrimer that is essential for the processivity of DNA polymerases by physically tethering the polymerases to DNA. In addition, PCNA acts as a loading platform for replication factors and proteins involved in cell cycle control and repair [19,20]. Importantly, these functions of PCNA are largely regulated by post-translational modifications at distinct lysine residues. Over the last two decades, PCNA ubiquitination and SUMOylation have been identified and extensively studied for their roles in DNA replication-coupled and post-replicative repair. Here, we will highlight the importance of PCNA ubiquitination and SUMOylation in the regulation of DDT pathways and how they act to maintain genome stability. Further, we will discuss how the misregulation of these pathways can lead to cellular transformation and tumorigenesis.

## 2. Bypass Replication

The first description of “bypass replication” (also known as post-replicative repair) dates back to 1968. Rupp and Howard-Flanders described discontinuities in newly synthesized DNA of nucleotide excision repair (NER) defective *E. coli* following UV radiation. These discontinuities were interpreted as single-stranded DNA gaps that were generated as the replication machinery stalled at pyrimidine dimers and resumed activity past the lesion [21]. The authors later proposed that these “dimer-gap structures” were resolved by a recombination-dependent process [22]. Shortly after, the concept of bypass replication was extended to mammalian cells and the mechanism of gap-filling was suggested to occur by *de novo* synthesis rather than recombination [23]. Since then, a significant amount of progress has been made toward understanding error-prone and error-free DNA damage bypass.

We now know that replication forks stall when they encounter damaged DNA, leading to the generation of long stretches of replication protein A (RPA)-coated single-stranded (ss) DNA [24,25,26,27,28,29]. These RPA-coated regions trigger the activation of two parallel pathways. The first is the ATR (Ataxia telangiectasia mutated and Rad3-related) checkpoint [29,30,31,32] and the second is the DDT pathway via the ubiquitination of PCNA at lysine 164 (K164) [17,18]. The principle mediator of DDT is the Rad6-Rad18 E2-E3 ubiquitin ligase complex [33,34,35,36,37]. However, the connection between PCNA ubiquitination and DNA damage remained enigmatic, until in 2002 Hoege et al., discovered that DNA damage induced mono-ubiquitination of PCNA occurs in a Rad6-Rad18 dependent manner [17]. Once mono-ubiquitinated, PCNA can be further modified by lysine 63 (K63)-linked poly-ubiquitin chains [17,37,38,39,40]. The nature of the ubiquitin modification determines which DDT pathway will be initiated, with mono-ubiquitination leading to TLS and poly-ubiquitination leading to TS [17,40,41].

Rad18 is the principal E3 ubiquitin ligase that mono-ubiquitinates PCNA, but several laboratories demonstrated the presence of residual mono-ubiquitinated PCNA in Rad18-deficient cells [42,43,44], indicating the existence of an alternative E3 ligase. RNF8 (ring finger protein 8) and CRL4(Cdt2) (Cullin-4-RING ligase (CRL4)-Ddb1-Cdt2) were later identified as two E3 ligases that can catalyze the mono-ubiquitination of PCNA. RNF8 in concert with the E2 conjugating enzyme, UbcH5c, readily mono-ubiquitinates PCNA in response to UV radiation and the alkylating agent, N-methyl-N′-nitro-N-nitrosoguanidine (MNNG) [45]. Furthermore, under unperturbed conditions, CRL4(Cdt2) catalyzes PCNA mono-ubiquitination and synergizes with Rad6-Rad18 to promote TLS [46]. Nevertheless, the contribution of RNF8 and CRL4(Cdt2) in the ubiquitination of PCNA is considered rather minor compared to Rad18.

### 2.1. Mono-Ubiquitination of PCNA: Error-Prone Lesion Bypass

Lesion bypass by TLS involves two critical events: polymerase switching and DNA extension past the lesion. High fidelity DNA polymerases, Pol ε and Pol δ, which facilitate leading and lagging strand synthesis, are replaced with low fidelity Y-family (TLS) polymerases that can use damaged DNA as templates and insert nucleotides opposite the lesions. These damage-tolerant DNA polymerases lack proofreading activity and have larger active sites that can accommodate bulky DNA adducts, thereby enabling replication to proceed directly past the lesion [15]. TLS polymerases are recruited to sites of DNA lesions through direct interaction with mono-ubiquitinated PCNA via their ubiquitin-binding zinc finger (UBZ; Pol η, Pol κ) and helical ubiquitin-binding motifs (UBM; Pol ι, Rev1) [47,48,49,50,51,52]. Additional interactions with PCNA occur through the canonical PIP (PCNA-interacting peptide) box found in Pol η, Pol ι, and Pol κ. Rev1, on the other hand, doesn’t have a PIP box, but interacts with PCNA through its N-terminal BRCA1 C-terminus (BRCT) domain and/or polymerase-associated domain (PAD) [53,54]. 

These interactions with PCNA stimulate the activity of TLS polymerases. In vitro, mono-ubiquitinated PCNA activates Pol η- and Rev1-dependent lesion bypass [55]. Mutations within the PIP box or ubiquitin-binding domain prevent damage-induced association of TLS polymerases with PCNA and their accumulation at replication factories [49,52]. Following nucleotide insertion across the lesion, the nascent strand—which is often mismatched—is extended by either Pol κ or by the B-family polymerase, Pol ζ (Rev3-Rev7) [56,57,58,59]. Therefore, TLS requires the sequential action of two DNA polymerases: one inserts a nucleotide opposite the lesion and the other extends from it.

A major unanswered question in the field is how polymerase switching occurs at the replication fork. One proposed model is the PCNA “toolbelt” [60,61,62,63]. Some of the most compelling studies consistent with the PCNA toolbelt model come from work in bacteria. The prokaryotic counterpart of PCNA, the β clamp, can simultaneously bind to both high-and low-fidelity DNA polymerases, suggesting that upon fork stalling, the β clamp can exchange the two polymerases, allowing for rapid DNA damage bypass [62,64]. Rev1 was previously shown to bind and organize TLS polymerases within a multi-protein complex [65,66,67]. The precise function of these interactions was not clear until—in 2016—Boehm et al., discovered that Rev1 functions as a molecular “bridge” between PCNA and different TLS polymerases to facilitate the switching event (Figure 1) [68].

Interestingly, in 2012, a number of groups discovered that the Rev3-Rev7 complex of Pol ζ binds to the Pol31 and Pol32 (p50 and p66 in humans) subunits of Pol δ, forming a four-subunit Pol ζ(4) complex [69,70,71]. The subunit sharing between Pol δ and Pol ζ was then proposed to be another mechanism to facilitate polymerase switching. This suggested that upon Pol δ stalling at a DNA lesion, the catalytic subunit of Pol δ, p125, dissociated and was replaced with Pol ζ (Figure 1). In the light of recent studies that determined that Pol δ can carry out both leading and lagging DNA synthesis, lesions on both parental DNA strands could be bypassed in this fashion [72,73].

To present a complete picture to the reader, we would like to mention that several reports provide evidence that TLS is not completely dependent on mono-ubiquitinated PCNA. Although some of these findings [74,75,76,77] have stirred some debate, collectively, they support the idea that TLS can operate in the absence of PCNA ubiquitination. First, in vitro reconstitution of PCNA ubiquitination from purified yeast proteins revealed that PCNA mono-ubiquitinated on all three monomers did not enhance the affinity or stimulate the activity of Pol η, Rev1, or Pol ζ [74]. Second, in budding yeast, mutations in the UBZ domain of Pol η did not affect its function in promoting TLS through UV-induced DNA lesions [75,76]. Later studies on human Pol η confirmed these results [77]. Third, budding yeast Pol ζ and Rev1 were activated independently of PCNA ubiquitination upon UV radiation in mutants of Pol δ [78]. Functional Pol δ continuously competed with TLS polymerases for the primer-template junction at the front side of PCNA. Thus, the primary function of PCNA ubiquitination could be to serve as a docking site for TLS polymerases to outcompete Pol δ for PCNA access in response to DNA damage. Fourth, in DT40 cells, PCNA ubiquitination and Rev1 had independent roles in the control of TLS. Rev1 played a central role in maintaining replication fork progression on damaged DNA, whereas PCNA ubiquitination was essential for filling in post-replicative gaps [79]. Finally, in 2011 Hendel et al., showed that a significant fraction (25–30%) of TLS occurred in *Pcna^K164R/K164R^* mutant mouse embryonic fibroblasts (MEFs) [80]. Together, these observations challenged the idea that TLS is a linear pathway in which Rad18 promotes PCNA mono-ubiquitination, which in turn stimulates the activity of TLS polymerases.

### 2.2. Poly-Ubiquitination of PCNA: Error-Free Lesion Bypass

Before the mechanisms underlying error-free lesion bypass were determined, extensive genetic studies in yeast identified Mms21, Ubc13, and Rad5 as central players in this pathway [37,38,39,81]. Rad5 interacts with Rad18, Ubc13, and PCNA [17,37]. These interactions are important for the recruitment of Ubc13-Mms2 to sites of DNA damage to mediate K63-linked poly-ubiquitin chain formation on mono-ubiquitinated PCNA to activate the error-free branch of lesion bypass [17,41,82]. Poly-ubiquitination of PCNA has also been observed in mammalian cells. In addition to Ubc13-Mms2, two Rad5 orthologs have been identified: helicase-like transcription factor (HLTF) and SNF2 histone linker PHD RING helicase (SHPRH) both of which promote PCNA poly-ubiquitination and function in the maintenance of genome stability [83,84,85,86]. However, MEFs lacking both HLTF and SHPRH still exhibit PCNA poly-ubiquitination, suggesting the existence of yet another E3 ligase [87].

Error-free lesion bypass has been proposed to occur via two distinct mechanisms: either by fork regression, or TS, both of which are dependent on PCNA poly-ubiquitination. Fork regression followed by nascent strand annealing and DNA synthesis is supported by in vitro studies that showed that both the helicase and annealing activities of Rad5 promote fork reversal [88]. Similar to Rad5, HLTF can also facilitate fork regression through its DNA translocase activity [89,90]. An alternative model for error-free lesion bypass relies on a TS mechanism involving the formation of sister chromatid junctions (SCJs). Evidence in support of this came from Vanoli et al., who showed that homologous recombination factors Rad55 and Rad57, together with Pol δ were required for the formation of damage-induced SCJs [91]. Furthermore, the ATPase and ubiquitin ligase activities of Rad5, and Hmo1 (high mobility group protein 1), a molecular DNA “bender” were also involved in the formation of SCJs [92,93]. These SCJs were then preferentially processed by the Sgs1 (slow growth suppressor 1)-Top3 (topoisomerase 3)-Rmi1 (RecQ mediated genome instability 1) complex [91,94].

Other players involved in TS are Pol α-Primase, Ctf4 and the cohesin complexes [95,96]. These proteins facilitate sister chromatid recombination by maintaining nascent sister chromatids in close proximity. In addition to its structural role, Pol α-Primase and Ctf4 function in repriming past the lesion to allow restart of replication and subsequent TS [95]. An alternative way to reprime DNA synthesis is catalyzed by PrimPol [97,98,99,100]. Moreover, recent genetic screens determined that the 9-1-1 (Rad9-Hus1-Rad1) clamp, outside of its role in checkpoint activation, together with Exo1 nuclease also contribute to TS [101]. It is likely that these error-free mechanisms are not mutually exclusive and that fork remodeling by regression enables template switching.

## 3. Enhancers of PCNA K164 Ubiquitination

It is worthwhile to note that TLS polymerases play a role in promoting PCNA mono-ubiquitination by serving as a scaffold, separate from their polymerization activity. The C-terminus of Pol η physically bridges Rad18 and PCNA to stimulate PCNA mono-ubiquitination [102]. The interaction between PCNA and different PIP boxes, specifically, PIP2 and PIP3 in Pol η, and PIP1 and PIP2 in Pol κ, are a key factor in modifying PCNA [103]. Since Rev1 has no PIP box, it promotes PCNA mono-ubiquitination by facilitating the accumulation of chromatin bound Rad18 [104]. Mono-ubiquitinated Rad18 (Rad18-Ub) prefers to dimerize with non-ubiquitinated Rad18. This dimerization inhibits the recruitment of non-ubiquitinated Rad18 to chromatin [105]. Rev1 competes with non-ubiquitinated Rad18 for the binding to Rad18-Ub, thereby facilitating the release of non-ubiquitinated Rad18 that can subsequently be re-recruited to chromatin to catalyze PCNA mono-ubiquitination [104].

Multiple other positive regulators of PCNA mono-ubiquitination have been identified. In yeast, depletion of Rfa1, a subunit of RPA, significantly reduced the modification. Further studies revealed that Rad18 was recruited to stalled forks by RPA to activate post-replicative repair [18]. Subsequent experiments in mammalian cells confirmed these results [106]. As mentioned before, the activation of ATR and DDT are thought to be parallel pathways. However, PCNA mono-ubiquitination following treatment with the carcinogen, benzo[a]pyrene dihydrodiol epoxide (BPDE), partially depends on ATR. Depletion of ATR led to a moderate reduction in BPDE-induced PCNA mono-ubiquitination [107]. In contrast, PCNA mono-ubiquitination in response to UV- and hydroxurea (HU) was ATR independent but still required ATR downstream substrates, Checkpoint kinase 1, Claspin, and Timeless [108].

Through affinity purification and mass spectrometry, SIVA1 (SIVA apoptosis-inducing factor 1), SART3 (squamous cell carcinoma antigen recognized by T Cells 3), and MAGE-A4 (melanoma associated antigen-A4) were identified as critical regulators of PCNA mono-ubiquitination in response to UV-induced DNA damage [109,110,111]. SIVA1 plays an important role in the extrinsic and intrinsic apoptotic pathways induced by the CD27 (cluster of differentiation 27) receptor, whereas SART3, a pre-mRNA splicing factor, functions in the recycling of the splicing machinery. Outside their known roles in apoptosis and mRNA splicing, SIVA1 and SART3 function as adaptors for Rad18 to bind to PCNA, and promote efficient PCNA ubiquitination in response to UV radiation [109,110]. Other factors, such as MAGE-A4 do not function as a bridge between Rad18 and PCNA, but instead directly bind to Rad18 to stabilize it. Not surprisingly, stabilization of Rad18 increases the ubiquitination of PCNA and the activation of TLS [111]. Additional regulators include NBS1 (Nibrin) and ZBTB1 (Zinc finger and BTB domain-containing 1), both of which are required for UV-induced Rad18 chromatin loading. NBS1 directly recruits Rad18 via a Rad6-like domain located in its C-terminus [112]. ZBTB1 promotes KAP-1 (Krüppel associated box (KRAB)-associated protein 1)-dependent chromatin relaxation at UV damage sites, thereby increasing the accessibility of Rad18 [113]. Lastly, a recent study has implicated BRCA1 (Breast cancer type 1 susceptibility protein) in both TLS and TS [114]. BRCA1 promotes the mono- and poly-ubiquitination of PCNA by actively recruiting RPA, Rad18, and HTLF to chromatin. Moreover, BRCA1 also directly recruits TLS polymerases, Pol η and Rev1, to perform lesion bypass [114].

## 4. New Readers of PCNA K164 Ubiquitination

As discussed above, mono-ubiquitinated PCNA can recruit different TLS polymerases allowing for lesion bypass and continued DNA replication. The conserved PIP box and ubiquitin-binding domain within these polymerases suggest that these two domains are important for the recognition of ubiquitinated PCNA. A previously uncharacterized human protein encoded by *C1orf124*, which contains a canonical PIP box, a C-terminal UBZ, and a N-terminal SprT-like domain is a reader of PCNA ubiquitination and a regulator of UV-induced DNA damage. This protein was eventually named Spartan (SprT-like domain at the N terminus, SPRTN) or DVC1 (DNA damage-targeting VCP adaptor C1orf124) [115,116,117,118,119,120]. All studies to date have come to a similar conclusion that both the PIP box and the UBZ domain of Spartan are important for its localization to sites of DNA damage.

The role of Spartan in the DNA damage response remains controversial. Some studies report that depletion of Spartan reduced the amount of chromatin-associated Rad18, Pol η, and ubiquitinated PCNA. Centore et al., interpreted Spartan as a positive feed-forward reader and writer of PCNA ubiquitination, whereas others proposed it to protect against PCNA de-ubiquitination [115,116]. Spartan has a DNA-dependent protease activity similar to that found in the yeast metalloprotease, Wss1 (weak suppressor of smt3) [121]. Spartan, like Wss1, facilitates replication by TLS after removing UV- and formaldehyde-induced DNA-protein crosslinks [121,122,123]. Collectively, these observations suggest that Spartan functions as a positive regulator of TLS following DNA damage. However, this conclusion has been challenged by reports that depletion of Spartan increased the association of Pol η with chromatin. Moreover, the conserved SHP box in Spartan interacts and recruits p97 to sites of DNA damage [119,120]. p97 segregase activity may facilitate removal of Pol η to prevent excessive TLS, implicating Spartan as a negative regulator of TLS.

As described above, multiple studies have focused on understanding the mechanism(s) of TLS. However, much less is known about the molecular underpinning(s) of TS following PCNA poly-ubiquitnation. The first protein to be functionally linked to poly-ubiquitinated PCNA was Mgs1 (maintenance of genome stability protein 1) in budding yeast [124]. Mgs1 was first identified as an AAA^+^ ATPase that contributes to genome stability both during unperturbed replication and in response to DNA damage [125,126]. Previous studies have shown that Mgs1 physically associates with PCNA [127] but it was not until 2012 when Saugar et al., showed that its N-terminal UBZ domain interacted with poly-ubiquitinated PCNA [124], similar to its mammalian homolog, Werner helicase interacting protein 1 [128,129]. Although Mgs1 has DNA-dependent ATPase and strand annealing activities, it is still unknown which of these activities are important for DNA damage bypass.

ZRANB3 (zinc finger, RAN-binding domain-containing 3), also known as AH2 (annealing helicase 2) is a binding partner of poly-ubiquitinated PCNA in mammalian cells [130,131,132]. The protein harbors a canonical PIP box and APIM (AlkB2 PCNA-interaction motif) domain that mediates its interaction with PCNA and a NPL4 zinc finger (NZF) motif that specifically recognizes K63-linked ubiquitin chains. ZRANB3 is transiently recruited to sites of stalled forks independently of PCNA, but subsequent binding to poly-ubiquitinated PCNA stabilizes ZRANB3 at the fork to (1) mediate global fork slowing [133], (2) promote replication fork restart via fork reversal (Figure 2), and (3) dissociate D-loop intermediates to limit recombination events [130,132]. ZRANB3 also acts as a structure-specific endonuclease that cleaves branched DNA structures, thus playing roles not only in DNA damage bypass but also in DNA repair (Figure 2) [131]. Together, these results demonstrate that ZRANB3 functions downstream of poly-ubiquitinated PCNA to promote active fork slowing and reversal to protect chromosome integrity in response to replication stress.

## 5. Timing of DDT: When and Where Does It Take Place?

There has been a long-standing debate on whether DNA damage bypass occurs in late S/G2 or is coupled to on-going DNA replication. Several studies in yeast strongly support that lesion bypass functions at replication forks [134,135,136,137,138,139]. However, multiple observations point to the fact that ssDNA gaps can persist into late S/G2: (1) the visualization of long ssDNA tracks behind forks in UV irradiated cells, and (2) the accumulation of small ssDNA gaps along replicated duplexes, suggestive of repriming events past a DNA lesion [134,140,141]. When the expression of Rad17, Rad18, Rad5, or TLS polymerases (Rad30, Rev3) was restricted to G2/M, no adverse effects on viability were observed, indicating that DNA damage bypass can be delayed and effectively function after bulk genome replication has been completed [101,136,137].

Despite the strong evidence for fork-independent gap filling, some studies suggest that DNA damage bypass, specifically error-free TS, is coupled to on-going replication forks. In budding yeast, the DNA helicase and branch migration activities of Rad5 are important for the reversal of stalled replication forks [88,142] and subsequent restart via sister chromatid recombination [92]. Furthermore, HLTF promotes replication fork reversal through its conserved HIRAN (HIP116 Rad5p N-terminal) domain. This domain can bind to 3′-ssDNA to recruit HLTF to stalled forks, and it directionally positions its double-stranded DNA translocase motor domain to mediate fork reversal [89,143,144]. Based on these reports, one has to conclude that DDT acts during S phase and also post-replicatively in G2.

## 6. De-Ubiquitination of PCNA

PCNA ubiquitination plays an essential role in the bypass of lesions encountered during DNA replication. The level of ubiquitinated PCNA must be strictly regulated to prevent mutagenesis by unscheduled recruitment of TLS polymerases. The de-ubiquitinating enzyme (DUB), USP1 (ubiquitin-specific peptidase 1) negatively regulates PCNA ubiquitination [44]. In the absence of DNA damage, USP1 exists in a catalytically active protein complex with UAF1 (USP1-associated factor 1) [145], which functions to regulate both the stability and the activity of USP1. Moreover, hELG1 (enhanced level of genomic instability 1), an alternative subunit of the RFC clamp loader complex, specifically directs the USP1-UAF1 dimer to PCNA for de-ubiquitination [146]. In response to UV irradiation, USP1 is inactivated through an autocleavage event, thus leading to the accumulation of mono-ubiquitinated PCNA and the activation of TLS [44,106].

USP7 (ubiquitin-specific peptidase 7) was identified as another DUB targeting mono-ubiquitinated PCNA. USP7 de-ubiquitinates PCNA in response to UV- and oxidative stress-induced DNA damage [147]. Recent work also shows that USP7 indirectly regulates the ubiquitination of PCNA by stabilizing Pol η or Rad18 [148,149]. Poly-ubiquitination of Pol η by Mdm2 [150,151,152] and auto-ubiquitination of Rad18 [105,153] targets these two proteins for proteasomal degradation. Pol η and Rad18 are stabilized by USP7-mediated de-ubiquitination, leading to efficient PCNA mono-ubiquitination [148,149]. A recent proteomic study has established that USP7 is present at the replisome as a SUMO de-ubiquitinase [154]. USP7 limits the ubiquitination of SUMOylated proteins at replication forks, preventing their extraction by the p97 segregase. The action of USP7 helps to maintain a SUMO-rich environment necessary for replication fork progression. Lastly, USP10, another DUB involved in the de-ubiquitination of PCNA, plays a crucial role in TLS termination. Following lesion bypass, mono-ubiquitinated PCNA is modified by interferon-stimulated gene 15 (ISG15), leading to the recruitment of USP10 and de-ubiquitination of PCNA. This in turn triggers the releases of Pol η from PCNA, allowing for the termination of TLS [155].

## 7. Other Modifications of PCNA

In addition to ubiquitination at K164, PCNA function is regulated by a variety of PTMs at other residues. These are summarized in Table 1 and discussed in detail below.

### 7.1. Ubiquitination at Alternate Lysines

PCNA ubiquitination at an alternate attachment site, K107, has been observed in budding yeast in response to defects in DNA ligase I [156,157]. The modification is dependent on the E2, Mms2 in conjugation with Ubc4, and the E3 ubiquitin ligase, Rad5. K107 ubiquitination is critical for the activation of the S phase checkpoint protein, Rad53, to elicit a robust damage response [156]. It has also been proposed that the ubiquitination of K107 functions as a nick sensor during Okazaki fragment maturation [157]. Moreover, in response to defects in Okazaki fragment maturation, PCNA is ubiquitinated at K242 [158]. Overexpression of a dominant negative mutant of the budding yeast flap endonuclease, Rad27, led to the ubiquitination of PCNA at K164 and a second site, K242. In the absence of K242 ubiquitination, the mutation rate decreased, which suggested that this modification promoted TLS.

Interestingly, a study in human cells using quantitative proteomics to profile ubiquitination, acetylation, and phosphorylation in response to UV and IR discovered that UV-induced ubiquitination of PCNA not only occurred at K164, but also at K117 [159]. To date, the function of K117 ubiquitination is still unknown. Taken together, these studies demonstrate that alternative attachment sites for ubiquitin exist on PCNA.

### 7.2. SUMOylation

SUMO modification of PCNA was first identified in budding yeast. PCNA is SUMOylated mainly at K164 and to a lesser extent at K127 by the SUMO E2-E3 complex Ubc9-Siz1 [17,40]. PCNA SUMOylation is a constitutive modification during S phase, and only occurs when PCNA is loaded onto DNA [160]. SUMOylated PCNA leads to the suppression of spontaneous homologous recombination (HR) through the recruitment of the anti-recombinogenic helicase, Srs2 [161,162]. Srs2 interacts with SUMO-PCNA through its C-terminal domain, which harbors a non-canonical PIP box and a SUMO-interaction motif (SIM) [163]. Both motifs are required for specific recognition of SUMO-PCNA. The SIM recognizes SUMO and the PIP-like motif binds to PCNA. However, structural and computational modeling revealed that SUMO associates with PCNA by simple tethering, allowing SUMO to adopt extended flexible conformations which do not interact with the surface of PCNA. These flexible conformations are required for proper Srs2 recruitment [164]. Two mechanisms have been proposed to explain how Srs2 suppresses HR at replication forks. One model proposed the active displacement of Rad51 nucleoprotein filaments preventing the initiation of recombination [161,162,165,166]. A more recent model suggested that Srs2 directly inhibits D-loop extension by Pol δ, limiting the extent of recombination [167,168]. The effect of Srs2-SUMO-PCNA can be viewed as a safety mechanism to prevent inappropriate damage bypass by HR, thereby facilitating ubiquitin-dependent TLS.

Interestingly, the ubiquitin ligase activity of Rad18 targeting PCNA was strongly enhanced by SUMO-PCNA. The stimulation was dependent on a SIM domain in Rad18 [169]. SUMOylated PCNA has been proposed to be the physiological substrate of Rad18, at least in yeast. These observations suggest that in response to DNA damage, Rad18 can directly ubiquitinate PCNA to promote DDT, without an intervening desumoylation step. Another protein that interacts with SUMO-PCNA is yeast Elg1. Elg1 interacts with SUMO-PCNA through three SIM motifs and a PIP-like motif located within its N-terminus [165]. Deletion of Elg1 causes the accumulation of SUMO-PCNA on chromatin, suggesting that the protein might act as a SUMO-PCNA unloader [170,171,172].

SUMOylation of PCNA at K164 has also been observed in *Xenopus laevis* egg extracts, chicken DT40 cells, and in mammalian cells, however, the abundance in vertebrate systems is significantly lower than in budding yeast [42,173,174,175]. In addition to K164, K254 was identified as a second SUMOylation site in human PCNA [174]. Similar to yeast, mammalian Ubc9 acts as the E2 conjugating enzyme. Surprisingly, SUMOylation of PCNA is not dependent the Siz1 orthologs, PIAS1-4. Instead, the interaction between PCNA and RFC appears to be necessary for PCNA SUMOylation [174]. Three functional homologs of Srs2 have been identified in human cells: RTEL1 (regulator of telomere elongation helicase 1) [176], FBH1 (F-box DNA helicase) [177,178] and PARI (PCNA-associated recombination inhibitor) [175,179]. RTEL1 is a Rad3-like helicase shown to inhibit recombination by disrupting D-loop structures, similar to Srs2. FBH1 and PARI both belong to the UvrD family of helicases and possess anti-recombinogenic activity [175,177,179]. Although both interact with PCNA, only PARI preferentially binds SUMOylated PCNA through its SIM domain [175]. FBH1 does not have a SIM domain, but has an additional F-box domain and a PIP degron. The PIP degron mediates the recruitment of CRL4(Cdt2) to promote degradation of FBH1 by the proteasome to allow for efficient recruitment of Pol η following UV radiation [178]. This suggests that FBH1 functions as a “molecular switch” that prevents HR to promote TLS during DNA replication. Future work will have to focus on better understanding the role of PARI and identifying other protein partners involved in regulating PCNA SUMOylation in human cells.

### 7.3. ISGylation

PCNA was recently identified as a target of ISG15. UBE1L-UBCH8-EFP (E1-E2-E3) specifically binds to mono-ubiquitinated PCNA and promotes its ISGylation at K164 and K168 following UV radiation [155]. Mutation of either residue (K164 or K168) prevented ISGylation of the other, indicating that ISGylation of one site affects the modification of the other site. Considering that PCNA ISGylation occurs after PCNA mono-ubiquitination, this suggests that ISG15 plays a role in the recovery from DNA damage. ISG-PCNA leads to the recruitment of USP10, which functions to de-ubiquitinate PCNA, triggering the release of Pol η from PCNA and facilitating the termination of TLS. PCNA is eventually de-ISGylated by UBP43, allowing for the reloading of replicative polymerases and the resumption of normal replication. It is currently unknown if other proteins are ISGylated following DNA damage and it will be interesting to see how this modification contributes to genome integrity.

### 7.4. Acetylation

In addition to ubiquitination and SUMOylation, acetylation plays multiple roles in regulating PCNA function. Acetylated PCNA has a higher affinity for Pol δ and Pol β compared to the deacetylated form, indicating that PCNA acetylation is important for normal DNA replication [180]. However, several studies have linked PCNA acetylation to its degradation following UV exposure [181,182]. In the absence of DNA damage, PCNA forms a complex with MTH2 (MutT homolog2), which stabilizes PCNA to allow for accurate replication fork progression. Mutational studies indicate that in response to UV radiation, PCNA is acetylated at K14, leading to its dissociation from MTH2 and degradation [181]. The histone acetyltransferase (HAT), p300 has been implicated in the acetylation of PCNA [180]. It was not until Cazzalini et al., identified that CREB-binding protein (CPB) in addition to p300 acetylates PCNA at K13, 14, 77, and 80 to promote the removal and degradation of chromatin-bound PCNA [182]. In the parasite, *Leishmania donovani*, the acetylation of PCNA by HAT3 also mediates its degradation. Interestingly, the acetylation of PCNA was found to precede PCNA mono-ubiquitination following UV radiation, suggesting that PCNA acetylation could facilitate the bypass of lesions encountered during replication [183]. Altogether, these studies implicate PCNA acetylation as a regulator of UV-induced PCNA turnover at replication forks.

A recent mass spectrometry analysis of purified yeast PCNA identified PCNA acetylation at K20 in response to DNA damage. K20 acetylation was dependent on Eco1 acetyltransferase, and the modification was important to stimulate cohesion-mediated HR and suppress the DNA damage sensitivity of DDT pathway mutants [184]. Moreover, K20 acetylation induced long-range conformational changes in PCNA that caused defects in the processivity of Pol δ. Altogether, this study demonstrates that Eco1-dependent acetylation functions at a lesion by favoring the removal of Pol δ from DNA and stimulating sister-chromatid recombination. Future studies will be important to clarify how the acetylation of PCNA at K20 rescues the damage sensitivity of DDT mutants.

### 7.5. Phosphorylation

Phosphorylation of PCNA is an important modification for downstream control of DNA replication and mismatch repair (MMR). PCNA phosphorylation on tyrosine 211 (Y211), by epidermal growth factor receptor (EGFR), is required for stabilizing PCNA on chromatin [185]. Additionally, the phosphorylation of PCNA at Y211 alters its interaction with MMR proteins, MutSα (MSH2-MSH6), MutSβ (MSH2-MSH3), and MutLα, thereby inhibiting MMR. Inhibition of MMR reduces the fidelity of DNA synthesis, leading to elevated levels of nucleotide misincorporation [186]. In the absence of Y211 phosphorylation, PCNA is poly-ubiquitinated at K164 by the E3 ubiquitin ligase Cullin4A, resulting in degradation of PCNA [187]. PCNA Y211 phosphorylation might represent an additional oncogenic target of EGFR, which promotes tumor development and progression through suppression of MMR and induction of error-prone DNA replication.

### 7.6. Methylation

Recent studies have demonstrated that PCNA methylation is a critical PTM controlling DNA replication. PCNA is methylated by the histone lysine methyltransferase, SETD8 on K248 and di-methylated by EZH2 at K10 [188,189]. Methylation at K10 and K248 is essential for the stability of PCNA. Methylation at K248 is specifically required for PCNA interaction with flap endonuclease 1 (FEN1) during Okazaki fragment maturation [188], whereas di-methylation at K10 is required for the binding of Pol δ to PCNA [189]. Both are critical for DNA replication.

## 8. Roles of PCNA Ubiquitination Outside of DDT

### 8.1. Lagging Strand Synthesis

The prevailing view is that PCNA ubiquitination at K164 and the activation of DDT pathways is induced by DNA damage. However, recent work has suggested that mutants with impaired replisome function activate DDT pathways in the absence of DNA damage [190,191,192]. Furthermore, PCNA ubiquitination has been observed during unperturbed replication in fission yeast [193], *Xenopus* egg extracts [173], and DT40 cells [43], suggesting that, in addition to controlling DDT, PCNA ubiquitination has a role during normal DNA replication. Strong evidence linking PCNA ubiquitination to DNA replication comes from a synthetic genetic array (SGA) analysis using a *PCNA^K164R^* mutant as a query against a library of budding yeast temperature-sensitive alleles. The SGA signature of the K164R allele mimicked that of many mutant alleles of genes involved in Okazaki fragment synthesis and maturation, connecting PCNA K164 to lagging strand synthesis [194]. A later study by Daigaku et al., found that PCNA ubiquitination stabilized chromatin-bound PCNA and affected the recruitment of Pol δ. In the absence of PCNA ubiquitination, DNA replication was slower and the frequency of ssDNA gaps increased [195]. These data suggest that PCNA ubiquitination acts to prolong the chromatin association of PCNA and Pol δ to allow for the completion of lagging strand synthesis via gap-filling.

### 8.2. Somatic Hypermutation, Class Switch Recombination, and Meiotic Recombination

Somatic hypermutation (SHM) is the process of diversifying the variable region of the immunoglobulin (Ig) genes by random nucleotide substitutions. Activation-induced deaminase (AID) initiates this process by cytosine deamination [196,197]. The resulting uracils are recognized by uracil glycosylase, UNG, or by mismatch repair proteins, MSH2-MSH6, resulting in mutations [198,199]. Contrary to their known roles in mediating high-fidelity repair, processing by UNG and MSH2-MSH6 leads to error-prone repair at the Ig variable regions. It has been suggested that the recruitment of error-prone TLS polymerases facilitates mutagenesis [200,201,202,203,204,205]. In 2006, Arawaka et al., first demonstrated that PCNA K164 has a role in SHM. A *Pcna^K164R^* mutation in the chicken DT40 B cell line caused a decrease in the number of Ig mutations, with C-to-G and G-to-C transversions being the most reduced [42]. However, a *Pcna^K164R^* mutation in mouse B cells showed a similar frequency of transitions and transversions as wildtype cells. Interestingly, when comparing the mutation patterns, A:T base pairs exhibited significantly less alterations in the *Pcna^K164R^* mutant B cells than in wildtype, suggesting the existence of a PCNA-dependent A:T mutator pathway [206,207]. In addition to SHM, PCNA K164 also plays a role in class-switch recombination (CSR). AID initiates CSR by deaminating cytosines in both the donor and acceptor switch regions, located upstream of each heavy chain constant region gene [196,208]. A *Pcna^K164R^* mutation impaired ex vivo class switch from IgM to IgG1 and IgG3 [207].

Interestingly, *Pcna^K164R^* mice develop normally but are sterile, suggesting that PCNA ubiquitination also plays a role in meiosis [206]. Indeed, *Pcna^K164R^* mice display a meiotic defect in early pachynema. Early meiotic progression in *Pcna^K164R^* mice appears normal, but meiotic nuclei were missing at stages beyond pachynema [207]. Altogether, PCNA ubiquitination at K164 plays critical roles in meiosis and the diversification of the Ig locus through SHM and CSR, however a detailed mechanism describing how PCNA controls these processes is still unknown.

## 9. Defects in DDT and Cancer Development

DDT pathways act as a fail-safe mechanism to allow for the completion of DNA replication. At the same time, they present a double-edged sword since one of the pathways, TLS, is intrinsically error-prone. At the expense of introducing mutations, TLS reduces the risk of genome rearrangements that come with prolonged replication fork stalling. Given this trade-off, tolerance pathways must be tightly regulated. Not surprisingly, the identification of mutations in TLS polymerases strongly predispose individuals to cancer development. Mutations in Pol η are known to cause xeroderma pigmentosum variant (XPV), an inherited disorder which is associated with an increased susceptibility to sunlight-induced skin cancers [209,210]. Compared to other XP groups that carry mutations in nucleotide excision repair, XPV patients have milder skin conditions. Because of this, XPV patients are not diagnosed until late in life, after they have accumulated UV-induced mutations that catalyze the development of multiple skin malignancies [211]. Moreover, mutations in the *POLH* gene, which encodes Pol η, have been identified in a small subset of breast and prostate cancers [212,213]. Mutations in other TLS polymerases, such as Pol ι and Pol κ have been linked to an increased predisposition to lung and prostate cancers [213,214,215,216,217]. In addition to these aforementioned mutations, overexpression of TLS polymerases also increases cancer risk [218,219,220]. A possible mechanism by which the overexpression of these polymerases promotes cancer progression was suggested by Garcia-Exposito et al., in 2016. Pol η plays a role in the alternative lengthening of telomeres (ALT). It manages replication stress at telomeres in ALT cells by maintaining telomere recombination at tolerable levels. Depletion of Pol η led to increased telomere instability and mitotic DNA synthesis [221].

Additionally, other proteins such as HLTF and SPRTN have been linked to cancer development. Several studies revealed reduced expression of *HLTF* caused by promoter hypermethylation in colon, esophageal, gastric, and uterine cancers, suggesting that *HLTF* silencing conferred a selective growth advantage [222]. On the other hand, high levels of HLTF protein expression were observed in early stages of an experimental model of estrogen induced kidney tumors, linking HLTF to the initial steps of carcinogenesis [223]. Furthermore, biallelic germline mutations in *SPRTN* have been linked to early onset hepatocellular carcinoma [224]. Altogether, this evidence suggests that a tightly regulated balance of DDT pathway components is required to preserve genome stability and prevent cellular transformation.

### Resistance to Platinum-Based Anti-Cancer Therapies

Several studies have correlated the upregulation of TLS with cisplatin resistance in multiple cancer models [225,226,227]. Rev3 and Rev1 were identified as key players in conferring cisplatin resistance in gliomas, lung adenocarcinomas, and B-cell lymphomas. Recently, the nucleosome remodeling factor, CHD4 was found to be a mediator of cisplatin response in *BRCA2* mutant cells [228]. Loss of CHD4 led to increased PCNA mono-ubiquitination and cisplatin resistance. Identification of a link between TLS and cisplatin resistance led to multiple studies in which DDT pathways were targeted for cancer therapy. One such study found that chemical inhibition of the Rad6-Rad18 pathway conferred cisplatin sensitivity to an intrinsically cisplatin-resistant triple negative breast cancer line [229,230]. Another study reported that transplantation of a DDT-proficient lymphoma cell line into cisplatin treated mice led to increased tumor burden and decreased survival compared to the transplant of DDT-deficient cells [231]. These studies highlight the importance of DDT pathways in cancer cell resistance to DNA damaging agents, making DDT inhibition a promising therapeutic option in the future.

## 10. Concluding Remarks

Studies over the past two decades have provided novel insights into the regulation of DDT pathways. Identification of new binding partners and functional modulators of PCNA has revealed the complexity of these pathways and the tight regulation that is necessary for the maintenance of genome stability. Current research efforts are focused on developing a deeper molecular understanding of each DDT pathway and the possible crosstalk between them. The field requires a more refined understanding of how polymerase switching occurs. Moreover, the full repertoire of proteins involved in DDT and their mechanistic roles are still unknown. Although recent studies have uncovered interesting links to platinum drug resistance in cancer, if and how DDT pathways contribute to tumorigenesis is largely obscure. We predict that future investigations into the relationship between DDT and cancer development could lead to innovative therapeutic approaches.

## Figures and Tables

**Figure 1 genes-10-00010-f001:**
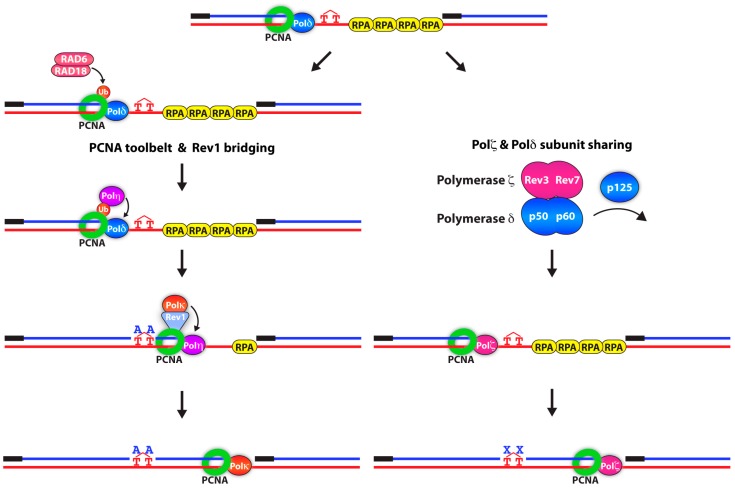
Two models for polymerase switching during translesion synthesis. In the PCNA toolbelt and Rev1 model, mono-ubiquitinated PCNA recruits the TLS polymerase, Pol η via its UBZ domain and replaces Pol δ for the insertion step. Following the insertion step across the lesion, Rev1 binds PCNA through its BRCT/PAD domain. Rev1 functions as a “bridge” to recruit Pol κ for the extension step. Pol κ replaces Pol η and extends the mismatched nascent DNA. In the Pol ζ and Pol δ subunit sharing model, upon Pol δ stalling, the catalytic subunit of Pol δ, p125, dissociates and is replaced by Rev3-Rev7 subunits of Pol ζ. This subunit sharing facilitates the switch from Pol δ to Pol ζ, thus bypassing the DNA lesion. X represents any nucleotide. If the lesion is replicated by Pol η, X is likely an “A”. However, other TLS polymerases such as Pol ζ can incorporate mismatched nucleotides. Abbreviations: UBZ: ubiquitin-binding zinc finger, BRCT: BRCA1 C-terminus, PAD: polymerase-associated domain.

**Figure 2 genes-10-00010-f002:**
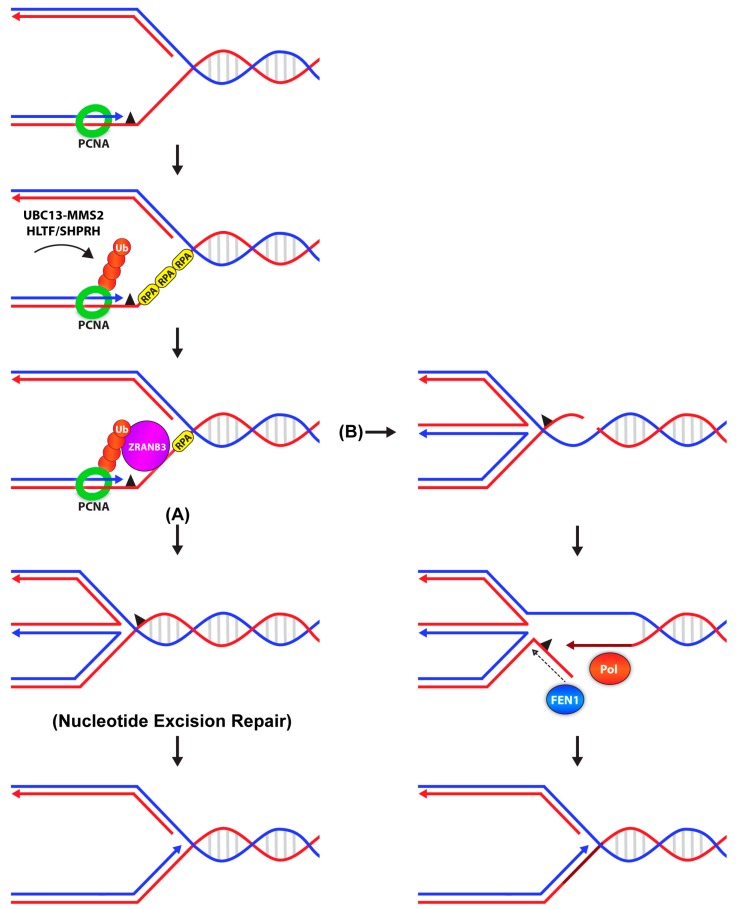
Error-free lesion bypass by ZRANB3. Upon replication fork stalling, PCNA is mono-ubiquitinated by the RAD6-RAD18 complex. K63-linked poly-ubiquitin chains are added by UBC13-MMS2 with the ubiquitin ligase, HLTF or SHPRH. ZRANB3 then binds to poly-ubiquitinated PCNA via its PIP, APIM and NZF motifs and (**A**) facilitates fork reversal. The remaining lesion will be processed by NER. (**B**) Alternatively, ZRANB3 acts as a structure-specific endonuclease and induces a DNA break, two nucleotides into the parental duplex, exposing a free 3′OH group. Fork reversal occurs to stabilize the fork while the free 3′OH is extended by DNA polymerase, displacing the lesion into a 5′ flap. The 5′ flap is then processed by FEN1, removing the lesion. DNA replication resumes following nick sealing and fork reversal. Model B is adapted from Weston et al., 2012, 26, 1558-1572. Abbreviations: HLTF: helicase-like transcription factor, SHPRH: SNF2 histone linker PHD RING helicase, ZRANB3: zinc finger, RAN-binding domain-containing 3, PIP: PCNA interacting peptide, APIM: AlkB3 PCNA-interaction motif, NZF: NPL4 zinc finger, NER: nucleotide excision repair, FEN1: Flap endonuclease 1.

**Table 1 genes-10-00010-t001:** Post-translational modifications of PCNA.

Modification	Target Site(s)	Species	Enzyme Modifiers	Readers	Function
Mono-ubiquitination	K164	Yeast and Human	Rad18 RNF8 CRL4^Cdt2^	Y-Family TLS polymerases	Promotes TLS
				Spartan	Promotes/Inhibits TLS?
	K117	Human	Unknown	Unknown	Backup for DDT pathway?
	K107	Yeast	Rad5	Unknown	Nick sensor for Okazaki fragment maturation
	K242	Yeast	Unknown	Unknown	Promotes TLS
Poly-ubiquitination	K164	Yeast and Human	Rad5 HLTF/SHPRH	Mgs1 ZRANB3	Promotes TS
SUMOylation	K164	Yeast and Human	Siz1 UBC9 (E2)	Srs2/PARI	Inhibits HR
	K254	Human	Unknown	Unknown	Unknown
	K127	Yeast	Unknown	Unknown	Unknown
ISGylation	K164, K168	Human	EFP	Unknown	Turns off TLS
Acetylation	K13, K14, K77, K80	Human	CPB/p300	Unknown	Promotes genome stability “PCNA turnover”
	K20	Yeast	Eco1	Unknown	Pol δ removal to stimulate sister chromatid recombination
Phosphorylation	Y211	Human	EGFR	Unknown	Protects against PCNA degradation and inhibits MutS binding
Methylation	K10 (di)	Human	EZH2	Unknown	DNA replication and cell proliferation
	K248	Human	SETD8	Unknown	DNA replication and cell proliferation

Listed are the modifications identified for PCNA in different model systems, their functional role, specific residues modified, and the enzyme responsible.

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
