# Peer review of "Mechanisms of DNA Damage Tolerance: Post-Translational Regulation of PCNA"

_genes, 2018, doi:10.3390/genes10010010_

Round 1

Reviewer 1 Report

This is a timely and important review article on a very important and active area of research. The authors do a nice job presenting a fair and comprehensive view of the field and the prevailing models. This manuscript should be a very helpful reference for researchers in adjacent fields as well as for researchers new to this field. As such, I feel that this manuscript is a worthy contribution to the field.

I have noted only one minor issue with the manuscript. On line 141, the authors cite a paper from the Prakash laboratory and state that the mono-ubiquitinated form of PCNA fails to stimulate TLS by polymerase eta and Rev1 in vitro. The issue is not as cut and dry as the authors make it seem. This was a single paper by the Prakash laboratory. Several papers from the Burgers laboratory and the Washington laboratory have shown the opposite result. I recommend that the authors either delete this point (which they can readily do by resting their argument on the other lines of evidence presented in the same paragraph) or acknowledge the controversial nature of this point.

Author Response

This is a timely and important review article on a very important and active area of research. The authors do a nice job presenting a fair and comprehensive view of the field and the prevailing models. This manuscript should be a very helpful reference for researchers in adjacent fields as well as for researchers new to this field. As such, I feel that this manuscript is a worthy contribution to the field.

I have noted only one minor issue with the manuscript. On line 141, the authors cite a paper from the Prakash laboratory and state that the mono-ubiquitinated form of PCNA fails to stimulate TLS by polymerase eta and Rev1 in vitro. The issue is not as cut and dry as the authors make it seem. This was a single paper by the Prakash laboratory. Several papers from the Burgers laboratory and the Washington laboratory have shown the opposite result. I recommend that the authors either delete this point (which they can readily do by resting their argument on the other lines of evidence presented in the same paragraph) or acknowledge the controversial nature of this point.

Answer: Reviewer 1 commented on the controversial nature of the paper we cited from the Prakash laboratory that demonstrated “PCNA monoubiquitinated on all three monomers did not enhance the affinity or stimulate the activity of Pol η, Rev1, or Pol ζ”. We have acknowledged this point (Lines 144-147 in the revised document).

Reviewer 2 Report

In this well-written review, Leung and colleagues comprehensively covered the history and current development of DNA damage tolerance (DDT) focusing on the role of PCNA modifications. The authors first introduced the “canonical” PCNA modifications – mono and poly ubiquitination. They explained how does ubiquitination regulates TLS and TS. Then the authors expanded into other posttranslational modifications such as SUMOylation and phosphorylation. Finally, the authors concluded with the physiological impact of PCNA modifications and its relationship with cancer treatments. Overall, the authors provided a very in-depth review of DDT in regarding to PCNA modifications. The authors even covered the studies showing that PCNA modifications are not absolutely needed for TLS activation.

I very much enjoyed reading this exceptional review and I believe that it provides an updated view of how PCNA post translational modification regulates the DDT. I recommend publication in Genes and would like the authors to consider adding some relevant information into this manuscript:

There have been a couple in vivo studies investigating the consequence of PCNA modifiers loss in mice. The authors may want to include some discussion on those works.

Author Response

In this well-written review, Leung and colleagues comprehensively covered the history and current development of DNA damage tolerance (DDT) focusing on the role of PCNA modifications. The authors first introduced the “canonical” PCNA modifications – mono and poly ubiquitination. They explained how does ubiquitination regulates TLS and TS. Then the authors expanded into other posttranslational modifications such as SUMOylation and phosphorylation. Finally, the authors concluded with the physiological impact of PCNA modifications and its relationship with cancer treatments. Overall, the authors provided a very in-depth review of DDT in regarding to PCNA modifications. The authors even covered the studies showing that PCNA modifications are not absolutely needed for TLS activation.

I very much enjoyed reading this exceptional review and I believe that it provides an updated view of how PCNA post translational modification regulates the DDT. I recommend publication in Genes and would like the authors to consider adding some relevant information into this manuscript:

There have been a couple in vivo studies investigating the consequence of PCNA modifiers loss in mice. The authors may want to include some discussion on those works.

Answer: Reviewer 2 suggested the addition of “in vivo studies investigating the consequence of PCNA modifier loss in mice”. We have cited the only two in vivo reports on PCNA modifier loss, “Nano-delivery of RAD6 / Translesion Synthesis Inhibitor SMI #9 for Triple-negative Breast Cancer Therapy” and “Precision cancer therapy: profiting from tumor specific defects in the DNA damage tolerance system” (ref. 231) in section 9, both of which were published in April and December of 2018. Most studies focused on the loss of PCNA modifiers have been performed in MEFs and not in vivo. If we missed a publication that this reviewer would like to see included it would be helpful to know which paper that is.

Reviewer 3 Report

Comments on Leung et al Mechanisms of DNA damage tolerance

This is a comprehensive historical survey of the last 20 years work investigating the roles of post translational modification of PCNA in response to DNA damage at the replication fork.

It is well written and readable. I have a few comments.

Section 2.1 might wish to include that TLS polymerase also lack proofreading capacity.

Figure 1 should have the TT lesion still present at the end on the right hand side too, with potential mutations incorporated on the replicated strand. I also thought that if pol delta was represented as a three subunit protein throughout, it would be simple to incorporate the subunit switch model on the DNA diagrams.

Line 139 I would disagree that “the model that monoubiquitinated PCNA promotes TLS has been challenged” by the observations listed next by the authors. It is clear from those studies that TLS is not completely dependent upon monoubiquitination, but I would say that it certainly promotes it. I draw the authors attention to disagreements surrounding ref77:  Sabbioneda S, et al. (2009) Ubiquitin-binding motif of human DNA polymerase η is required for correct localization. Proc Natl Acad Sci USA doi:10.1073/pnas.0812744106 and  Reply to Sabbioneda et al.: Role of ubiquitin-binding motif of human DNA polymerase η in translesion synthesis. Narottam Acharya, et al. (2009) 

PNAS February 24, 2009 106 (8) E21; https://doi.org/10.1073/pnas.0900176106.

Section 2.2 from line 181 would be more comprehensive if it had included mention of the recently identified  Primpol (citing work from the Huang, Mendez and Doherty groups) which although not a PCNA interactor is clearly vital for TLS and repriming at stalled forks – how it’s activity is integrated with the PCNA-dependent pathways remains of key interest.

Section 4 from line 241 where citing refs 117 and 118 the paper “Metalloprotease SPRTN/DVC1 Orchestrates Replication-Coupled DNA-Protein Crosslink Repair.” Vaz B, et al.

Mol Cell. 2016 Nov 17;64(4):704-719. doi: 10.1016/j.molcel.2016.09.032. Epub 2016 Oct 27. Should be included.

Section 5 from line 292 should also include consideration of the paper “PCNA ubiquitination and REV1 define temporally distinct mechanisms for controlling translesion synthesis in the avian cell line DT40.” Edmunds CE, Simpson LJ, Sale JE. Mol Cell. 2008 May 23;30(4):519-29. doi: 10.1016/j.molcel.2008.03.024.

Section 6, while initial papers did show hElg1 to be involved in deubiqitination it has subsequently become clear that it’s main role is in PCNA unloading (see papers by Donaldson Kubota and Kupiec groups).

In summary, this review is comprehensive and clear. There have not been any dramatic paradigm shifts regarding the roles of PCNA modification in recent years, and so this represents a solid overview of a mature field.

Author Response

This is a comprehensive historical survey of the last 20 years work investigating the roles of post translational modification of PCNA in response to DNA damage at the replication fork.

It is well written and readable. I have a few comments.

Section 2.1 might wish to include that TLS polymerase also lack proofreading capacity.

Answer: Reviewer 3 suggested to include that TLS polymerases lack proofreading capacity in section 2.1 “Monoubiquitination of PCNA: Error-prone lesion bypass”. We have included this point (Line 97 in the revised document).

Figure 1 should have the TT lesion still present at the end on the right hand side too, with potential mutations incorporated on the replicated strand. I also thought that if pol delta was represented as a three subunit protein throughout, it would be simple to incorporate the subunit switch model on the DNA diagrams.

Answer: Reviewer 3 commented “Figure 1 should have the TT lesion still present at the end on the right-hand side”. Figure 1 has been changed to show the TT lesion with potential mismatches incorporated represented as “x” on the replicated strand. The figure legend has been changed to include “X represents any nucleotide. If the lesion is replicated by Pol η, X is likely an “A”. However, other TLS polymerases such as Pol ζ can incorporate mismatched nucleotides”. Reviewer 3 also suggested to have Pol δ represented as a three-subunit protein throughout Figure 1. We considered this change, but in our view the subunit sharing between Pol δ and Pol ζ is better emphasized in the original version. Therefore, we opted against making this change.

Line 139 I would disagree that “the model that monoubiquitinated PCNA promotes TLS has been challenged” by the observations listed next by the authors. It is clear from those studies that TLS is not completely dependent upon monoubiquitination, but I would say that it certainly promotes it. I draw the authors attention to disagreements surrounding ref77:  Sabbioneda S, et al. (2009) Ubiquitin-binding motif of human DNA polymerase η is required for correct localization. Proc Natl Acad Sci USA doi:10.1073/pnas.0812744106 and  Reply to Sabbioneda et al.: Role of ubiquitin-binding motif of human DNA polymerase η in translesion synthesis. Narottam Acharya, et al. (2009) 

PNAS February 24, 2009 106 (8) E21; https://doi.org/10.1073/pnas.0900176106.

Answer: Reviewer 3 disagreed with the statement “the model that monoubiquitinated PCNA promotes TLS has been challenged”. We have altered the text to tone down the statement and have acknowledged the controversy surrounding ref. 77 (Lines 144-147 in the revised document).

Section 2.2 from line 181 would be more comprehensive if it had included mention of the recently identified  Primpol (citing work from the Huang, Mendez and Doherty groups) which although not a PCNA interactor is clearly vital for TLS and repriming at stalled forks – how it’s activity is integrated with the PCNA-dependent pathways remains of key interest.

Answer: Reviewer 3 suggested that section 2.2 would be more comprehensive with the addition of a reference to PrimPol. We agree with this suggestion and have added the text: “An alternative way to reprime DNA synthesis is catalyzed by PrimPol (Lines 192-193 in revised document).

Section 4 from line 241 where citing refs 117 and 118 the paper “Metalloprotease SPRTN/DVC1 Orchestrates Replication-Coupled DNA-Protein Crosslink Repair.” Vaz B, et al.

Mol Cell. 2016 Nov 17;64(4):704-719. doi: 10.1016/j.molcel.2016.09.032. Epub 2016 Oct 27. Should be included.

Answer: Reviewer 3 suggested including the paper “Metalloprotease SPRTN/DVC1 Orchestrates Replication-Coupled DNA-protein Crosslink Repair” in section 4. We agree with this suggestion and have added this reference ( number 123).

Section 5 from line 292 should also include consideration of the paper “PCNA ubiquitination and REV1 define temporally distinct mechanisms for controlling translesion synthesis in the avian cell line DT40.” Edmunds CE, Simpson LJ, Sale JE. Mol Cell. 2008 May 23;30(4):519-29. doi: 10.1016/j.molcel.2008.03.024.

Answer: Reviewer 3 suggested including the paper “PCNA ubiquitination and REV1 define temporally distinct mechansims for controlling translesion synthesis in the avian cell line DT40”, in section 5. This paper had already been referenced in section 2.1. We believe this is a better fit. We moved it to section 5 only to discover that it completely disrupts the flow. So we moved it back where it was originally.

Section 6, while initial papers did show hElg1 to be involved in deubiqitination it has subsequently become clear that it’s main role is in PCNA unloading (see papers by Donaldson Kubota and Kupiec groups).

Answer: Reviewer 3 commented about section 6, “while initial papers did show hElg1 to be involved in deubiquitination, it has subsequently become clear that its main role is in PCNA unloading”. We agree with the reviewer. However, we are talking specifically about human Elg1, and a role for PCNA unloading has not been demonstrated for human Elg1. The paragraph is about the regulation of deubiquitination, not PCNA unloading. Therefore, we did not add references to the work in yeast.

In summary, this review is comprehensive and clear. There have not been any dramatic paradigm shifts regarding the roles of PCNA modification in recent years, and so this represents a solid overview of a mature field.